# Modeling Elastomer Compression: Exploring Ten Constitutive Equations

**DOI:** 10.3390/ma16114121

**Published:** 2023-05-31

**Authors:** Stanisław Kut, Grażyna Ryzińska

**Affiliations:** Department of Materials Forming and Processing, Rzeszow University of Technology, al. Powstańców Warszawy 8, 35-959 Rzeszów, Poland

**Keywords:** elastomer, elastomer testing, constitutive equations, experiment, FEM simulation

## Abstract

This paper presents the results of research aimed at assessing the effectiveness of ten selected constitutive equations for hyperelastic bodies in numerical modeling of the first compression load cycle of a polyurethane elastomer with a hardness of 90 Sh A depending on the methodology for determining the material constants in the constitutive equations. An analysis was carried out for four variants for determining the constants in the constitutive equations. In three variants, the material constants were determined on the basis of a single material test, i.e., the most popular and available in engineering practice, the uniaxial tensile test (variant I), the biaxial tensile test (variant II) and the tensile test in a plane strain (variant III). In variant IV, the constants in the constitutive equations were determined on the basis of all three above material tests. The accuracy of the obtained results was verified experimentally. It has been shown that, in the case of variant I, the modeling results depend to the greatest extent on the type of constitutive equation used. Therefore, in this case it is very important to choose the right equation. Taking into account all the investigated constitutive equations, the second variant for determining the material constants turned out to be the most advantageous.

## 1. Introduction

An elastomer is a polymer that exhibits a nonlinear, elastic behavior under load. The term elastomer is often used to refer to materials that exhibit rubber-like behavior. However, elastomers are a wide group of materials. 

Elastomers are unique to polymers and exhibit extraordinary reversible extension with low hysteresis and minimal permanent set. They are the ideal polymers relieved of molecular interactions, crystallinity and chain rigidity constraints. The common elastomers have a characteristic low modulus, although with poor abrasion and chemical resistance. Elastic strain may be due to chemical bond stretching, bond angle deformation or crystal structure deformation. In an elastomer under strain, bonds are not elongated and bond angles not deformed. Stretch of an elastomer depends upon rotations about bonds, that is, changes to dihedral angles. An unstrained elastomer will exist in a random coil structure. As strain is increased, the molecules will uncoil to the limiting linear structure. Therefore, to be an elastomer, a substance essentially must consist of macromolecules. Large strain requires very long molecules so that uncoiling can be considerable. Formation of an unstrained random coil means that the elastomer must be non-crystalline since any regular crystal structures will be unable to contribute to the elastomeric properties [1,2].

In general, the behavior of rubber-like materials can be described as a function of strain energy density. Therefore, many attempts have been made to theoretically recreate the stress–strain curves obtained from experiments on the deformation of elastomeric materials. However, since the deformation capacity of these types of materials is high, and their behavior varies significantly from one type to another, it is difficult to determine a strain energy density function that adequately represents the stress–strain relationship from the experiments. Hence the need for research in this area.

Elastomers are widely used, including in the machinery and transport industry. They are also used in the textile, chemical, pharmaceutical and food industries. They are used as structural elements, elastomeric bearings, tires, medical devices and vibration isolators. In practice, elastomers are used in almost every area of our lives. Therefore, there is a great interest in the most accurate prediction of the behavior of these materials subjected to different stress states in specific applications already at the FEM design stage [3]. 

To describe the behavior of elastomers, a number of known material models are used, which, as a rule, should meet the criteria of accurate mapping of the entire deformation, insensitivity to changes in the stress state and changes in the deformation mode, the number of parameters (constants) should not be large and the equation should be easy to implement. 

In order to correctly describe the behavior of an elastomeric material using a suitable model, it must be taken into account that the material is typically subjected to a complex state of stress. This determines the stress conditions in which tests should be carried out to determine the constants in the constitutive equations [4,5]. In other words, the load pattern (state of stress and strain) in the material test should be as close as possible to the load pattern in real conditions of the analyzed structures or processes.

Attempts to compare the operation of selected material models (Mooney–Rivlin, Yeoh) are shown in [6], where only the compression test was used to determine the material constants.

It was proved that the Yeoh three-parameter model showed a better fit with the test data than the one-parameter formula. The two-parameter Mooney–Rivlin model showed a good fit with the experimental data, therefore it was indicated as promising for applications with other types of elastomers, including rubber. Comparing both of the above models, the Yeoh model turned out to be more accurate. On the other hand, the high accuracy of the Mooney–Rivlin model was demonstrated in the work on non-pneumatic safety tires [7]. 

However, considering the estimated error, the Yeoh three-parameter material model using the absolute error calculation for the curve fitting process shows the best accuracy. Four models were compared in [8]: Ogden, Neo-Hookean, Yeoh and Mooney–Rivlin.

The constants were determined in the uniaxial tensile test. The fit of the Ogden and Yeoh models for small and large deformations with the data obtained from the uniaxial tensile test turned out to be better compared to the Neo-Hookean and Mooney–Rivlin models, which showed low accuracy. In the works [8,9] using three tests: uniaxial tensile, pure shear test and the equibiaxial extension test, it was shown that the Arruda–Boyce model for large deformations provides accurate data over the entire strain range for all stress states. The Monney–Rivlin model is not applicable to large deformations. The Gent model also shows good accuracy of results in pure shear, uniaxial tensile, uniaxial compression and flat compression tests. 

Depending on the number and type of tests selected for determining the material constants, the equations present a different effectiveness and accuracy. Therefore, it is worth paying attention to the correct selection of the number and type of tests, which will allow them to be adapted to specific applications. 

The aim of this study was to evaluate the effectiveness of ten selected constitutive equations for hyperelastic bodies in the numerical modeling of the first compression load cycle for a polyurethane elastomer with a hardness of 90 Sh A depending on the methodology used for determining the material constants in the constitutive equations. Four variants for determining the material constants in the constitutive equations were used. 

In variant I, the material constants in the individual constitutive equations were determined solely on the basis of the uniaxial tensile test. In variant II, the material constants were determined only on the basis of the biaxial tensile test, while in variant III, they were determined based on a plane strain tensile test. However, in variant IV, the material constants were determined on the basis of three material tests, i.e., uniaxial and biaxial tensile tests and a plane strain tensile test.

## 2. Selected Constitutive Models

In engineering practice, only parameters such as the Shore A hardness and linear elastic modulus are often known for individual elastomers. In a limited range of deformation, the behavior of the elastomer may be close to linear, hence the popularity of classic simple models such as Neo-Hookean or Mooney–Rivlin model. However, a wider range of deformations and the influence of other parameters require the use of more advanced models. Therefore, constitutive models ranging from the simplest to the more complex and demanding, depending on needs and computational capabilities, were selected for the research. 

### 2.1. Neo-Hookean Model 

The Neo-Hookean model is a hyperelastic material model that can be used to predict the stress–strain behavior of materials, and the model is similar to Hooke’s law. The neo-Hookean model is one of the simple models describing the behavior of a material in the initial stage of deformation, and the deformation energy density function (for an incompressible material) is as follows:(1)$$W_{dev}=C_{10}\left(I_{1}-3\right)$$
where *W_dev_* is the deviatoric third-order deformation form strain energy function, *C*_10_ is a material constant and *I*_1_ is the first invariant of the left Cauchy–Green deformation tensor.

The Neo-Hookean model is based on the statistical thermodynamics of cross-linked polymer chains. Cross-linked polymers work in a manner consistent with the Neo-Hookean model in terms of linear deformations. However, at some point the polymer chains will be stretched to the maximum point that covalent cross-links will allow, and this will dramatically increase the material’s modulus of elasticity. It is well known that the model of a Neo-Hookean material does not accurately predict the behavior of an elastomer at high strain.

### 2.2. Mooney–Rivlin Model (Two-Parameter)

The Mooney–Rivlin model, which was introduced by Melvin Mooney and Ronald Rivlin, is a hyperelastic material model in which the strain energy density function “*W*” is a linear combination of two invariants of the left Cauchy–Green deformation tensor. Basic assumptions:The elastomer is incompressible and isotropic (in an unstrained state).The strain energy function must depend on even powers of λ_i_ (λ_i_—principal stretch ratio).

The strain energy density function for a Mooney–Rivlin (two-parameter) material is as follows:(2)$$W_{dev}=C_{10}\left(I_{1}-3\right)+C_{01}\left(I_{2}-3\right)$$

### 2.3. Mooney–Rivlin Model (Three-Parameter)

The strain energy density function for a Mooney–Rivlin (three-parameter) material is as follows:(3)$$W_{dev}=C_{10}\left(I_{1}-3\right)+C_{01}\left(I_{2}-3\right)+C_{11}\left(I_{1}^{2}-9\right)$$
where *C*_10_, *C*_01_ and *C*_11_ are experimentally determined material constants, and *I*_1_ and *I*_2_ are the first and the second invariants of the deviatoric component of the left Cauchy–Green deformation tensor.
(4)$$I_{1}=\lambda_{1}^{2}+\lambda_{2}^{2}+\lambda_{3}^{2}$$
(5)$$I_{2}=\lambda_{1}^{2}\lambda_{2}^{2}+\lambda_{2}^{2}\lambda_{3}^{2}+\lambda_{3}^{2}\lambda_{1}^{2}$$
(6)$$I_{3}=\lambda_{1}^{2}\lambda_{2}^{2}\lambda_{3}^{2}$$

Incompressibility implies that *I*_3_ = 1.

Material constants of the Mooney–Rivlin model are related to the initial shear modulus G and G can be expressed as follows:(7)$$G=2\left(C_{10}+C_{01}\right)$$

Despite the known limitations in the description of individual stress states, it is known that the Mooney–Rivlin model can be used for various structural elements with local deformation values up to about 200% [10,11,12,13]. 

### 2.4. Signiorini Model 

The Signiorini, Yeoh and James Green Simpson [14,15,16,17] models were proposed as variants of the Mooney–Rivlin model. The strain energy density function (assuming that the material is incompressible) is as follows [13]:(8)$$W_{dev}=C_{10}\left(I_{1}-3\right)+C_{01}\left(I_{2}-3\right)+C_{20}{\left(I_{1}-3\right)}^{2}$$

### 2.5. Yeoh Model 

The Yeoh form of the strain energy density function can be presented as follows [13,14,15]:(9)$$W_{dev}=C_{10}\left(I_{1}-3\right)+C_{20}{\left(I_{1}-3\right)}^{2}+C_{30}{\left(I_{1}-3\right)}^{3}$$

### 2.6. James, Green and Simpson Model

The James, Green and Simpson form [13,16,17] is as follows:(10)$$W_{dev}=C_{10}\left(I_{1}-3\right)+C_{01}\left(I_{2}-3\right)+C_{11}\left(I_{1}-3\right)\;\left(I_{2}-3\right)+C_{20}{\left(I_{1}-3\right)}^{2}+C_{30}{\left(I_{1}-3\right)}^{3}$$

### 2.7. Ogden Model

The Ogden model is a hyperelastic material model that can be used to predict a nonlinear stress–strain relationship for materials such as rubber or other types of elastomer. The Ogden model was introduced by Ogden in 1972, and the strain energy density function is as follows [13,18,19]:(11)$$W_{dev}=\sum_{k=1}^{N}\frac{\mu_{k}}{\alpha_{k}}({\bar{\lambda }}_{1}^{\alpha_{k}}+{\bar{\lambda }}_{2}^{\alpha_{k}}+{\bar{\lambda }}_{3}^{\alpha_{k}}-3)$$
where ${\bar{\lambda }}_{i}^{\alpha_{k}}$ is the deviatoric stretch ratio, μ_k_ and α_k_ are empirically determined material constants (material constants obtained from curve fitting of experimental data. This function is available in Mentat). If the volumetric modulus is not given, the material is considered to be practically incompressible. This model differs from the Mooney model in several respects. The Mooney material model refers to the invariants of the right or left Cauchy–Green strain tensor and implicitly assumes that the material is incompressible. The Ogden formula refers to the Cauchy–Green deformation value, and the presence of a volumetric modulus implies a certain compressibility. The Ogden model is a frequently used model to analyze rubber components such as O-rings and seals. Moreover, it has the advantage that the test data can be used directly. 

### 2.8. Arruda–Boyce Model

In the Arruda–Boyce strain energy model, the underlying molecular structure of the elastomer is represented by an eight-chain model to simulate the non-Gaussian behavior of individual chains in the network. The two parameters nkθ, and N (n is the chain density, k is the Boltzmann constant, θ is the temperature and N is the number of statistical links of length one in the chain between chemical cross-links) representing the initial modules and limiting chain extensibility are related to the molecular chain orientation, thus representing the physics of the network deformation.

In most models describing elastomer deformation, the strain energy function constructed by fitting the experimental data obtained from one strain state cannot be effectively used to describe another, altered, strain state. The Arruda–Boyce model improves on this inconvenience and is unique in this regard, as standard tensile test data provide sufficient accuracy for many other strain states. 

The model is constructed by using an eight-chain network [13,20].

Consider a cube of dimension $\alpha_{0}$ with an unstretched network including eight chains of length $r_{0}=\sqrt{Nl}$, where the fully extended chain has an approximate length of *Nl*. A chain vector from the center of the cube to a corner can be expressed as:(12)$$C_{1}=\frac{\alpha_{0}}{2}\lambda_{1}i+\frac{\alpha_{0}}{2}\lambda_{2}j+\frac{\alpha_{0}}{2}\lambda_{3}k$$

Using geometrical considerations, the chain vector length can be written as:(13)$$r_{chain}=\frac{1}{\sqrt{3}}\sqrt{Nl}{\left(\lambda_{1}^{2}+\lambda_{2}^{2}+\lambda_{3}^{2}\right)}^{1/2}$$
and:(14)$$\lambda_{chain}=\frac{r_{chain}}{r_{0}}=\frac{1}{\sqrt{3}}{\left(I_{1}\right)}^{1/2}$$

Using statistical mechanics considerations, the work of deformation is proportional to the entropy change on stretching the chains from the unstretched state and may be written in terms of the chain length as:(15)$$W=nk\theta N\left(\frac{r_{chain}}{Nl}\beta +ln\frac{\beta }{sinh\beta }\right)-\theta \hat{C}$$
where *n* is the chain density and $\hat{C}$ is a constant. β is an inverse Langevin function that correctly accounts for the limiting chain extensibility and is defined as:(16)$$\beta =L^{-1}\left(\frac{r_{chain}}{Nl}\right)$$

With the above equations, the Arruda–Boyce model can be written:(17)$$W_{dev}=nk\theta \left(\frac{1}{2}\left(I_{1}-3\right)+\frac{1}{20N}\left(I_{1}^{2}-9\right)+\frac{11}{1050N^{2}}\left(I_{1}^{3}-27\right)\;\;\;\;\;\;\;\;+\frac{19}{7000N^{3}}\left(I_{1}^{4}-81\right)+\frac{519}{673750N^{4}}\left(I_{1}^{5}-243\right)\right)$$

### 2.9. Gent Model

Using the notion of limiting chain extensibility, Gent [13,21] proposed the following constitutive relation:(18)$$W_{dev}=-\frac{E}{6}\left(I_{m}-3\right)\log\left(1-\frac{I_{1}-3}{I_{m}-3}\right)$$

The constant *EI_m_* is independent of the molecular length and, hence, of the degree of cross-linking. The model is attractive due to its simplicity, but yet captures the main behavior of a network of extensible molecules over the entire range of possible strains.

### 2.10. Marlow Model

The Marlow model is directly based on experimental data. It assumes that the strain energy function can be given as a function of the first invariant only. The form of this function is not defined explicitly, but its derivatives with respect to the first invariant, which are needed to compute stress and stiffness, are evaluated directly from test data. This means that no curve fitting is required to obtain material parameters of a strain energy function and that the supplied test data can be reproduced exactly by the material model. The test data can be obtained from either a uniaxial tensile test, a biaxial tension test or a pure shear (planar tension) test. Although the model reproduces the test data exactly, other modes of deformation may only be reproduced in an approximate sense.

The Marlow model can be used to simulate incompressible or slightly compressible behavior. If compressibility is to be included, some information about a volumetric strain energy function must be entered [13,14]. 

The initial shear modulus is estimated from the test data. The strain energy function is assumed to be the sum of a deviatoric *W*(*I*_1_) and a volumetric part *U*(*J*):(19)$$W=W\left(I_{1}\right)+U\left(J\right)$$

## 3. Methods

The achievement of the research goal required the development of an original research plan, which is schematically presented in Figure 1. The research plan included a part related to experimental research as well as a part related to modeling with the use of nonlinear FEM. The experimental part of the research concerned the material tests necessary to determine the constants in the constitutive equations and the verifying compression test. In order to determine the material constants via four variants, the following tests were carried out: uniaxial tensile test, biaxial tensile test and plane strain tensile test [13,22,23].

The material tests and the conditions for their implementation are described below. In order to verify the results of FEM modeling, experimental compression tests of the elastomeric cylinder were carried out. The force–displacement characteristic of the tool was determined, which was the basis for the verification of the FEM results with the experimental results. An important stage of the research was the development of the stress–strain curve for the first load cycle of the sample on the basis of individual material tests. Then, on the basis of these runs, the material constants in the constitutive equations that were selected for the research were calculated. The knowledge of the material constants for individual elastomer models was the basis for starting the research related to FEM modeling. A numerical model was built for each test as in the experiment, and then FEM calculations for compression of the elastomeric cylinder were performed using individual constitutive models for the first load cycle of the elastomeric specimen. Detailed information on the experiment and numerical modeling is included in the further part of the work.

## 4. Experimental Work

Due to the specificity of the elastomer depending on the method of loading, it is recommended [3,24,25,26,27,28] that the material constants in the constitutive equations of elastomers should be determined based on three material tests, i.e., uniaxial tensile test, biaxial tensile test and plane strain tensile test or pure shear test.

The uniaxial tensile test is the most well-known and popular material test in engineering practice. Due to the method of loading, it can be performed on a typical testing machine. In the case of the biaxial tensile test, an image of the behavior of an elastomeric sample in a plane stress state can be obtained [28,29,30,31]. It should be noted that, for incompressible materials, the deformation in the biaxial tensile test is the same as in the uniaxial compression test without friction. For this reason, this test determines well the behavior of rubber-like materials when subjected to compression. However, due to the inability to eliminate the frictional forces, this test cannot be replaced with a simpler uniaxial compression test. On the other hand, the tensile test is recommended [32] when determining solid rubber-like materials instead of the pure shear test, which in practice is more difficult to carry out correctly, especially for large deformations. This approach is justified because, in the middle part of an incompressible elastomeric material when tensile in a plane deformation state (no deformation along the width of the sample), the deformation pattern is the same as during the pure shear test. Due to the fact that the elastomer is almost incompressible, in the sample there is a pure shear state at an angle of 45 degrees to the tensile direction [32,33,34]. Therefore, in engineering practice, when determining material constants, the most commonly used is the plane strain tensile test as an alternative to the pure shear test. It is recommended that the sample for this test should be at least ten times wider than its length [28,35]. The fulfillment of this condition is necessary to obtain the required plane strain in the central part of the sample during uniaxial tensile stress, i.e., without deformations in the width of the sample. 

### 4.1. Uniaxial Tensile Test

The uniaxial tensile tests were carried out on the ZWICK/ROELL Z030 testing machine, which was equipped with a fully automatic multi-strain gauge with a variable measurement base and the testXpert II software. A view of the stand and the sample during the tensile test is shown in Figure 2. The shape and dimensions of the test specimens are shown in Figure 3 and Figure 4.

Uniaxial tensile tests were carried out in the range of deformation ε = 0–60%. In the case of a uniaxial tensile test, the deformation can be calculated as:(20)$$\varepsilon =\frac{\Delta l}{l_{0}}=\frac{l_{\max}-l_{0}}{l_{0}}$$
where: Δ*l*—elongation in uniaxial tensile test, *l*_max_—maximum length, *l*_0_—initial length, *ε*—engineering strain.

Each of the three samples was subjected to a full cycle of loading and unloading with constant elongation of the sample Δ*l* = 30 mm (*l*_0_ = 50 mm, thickness *g*_0_ = 4 mm). During each tensile test, the force course and the elongation of the sample in the measuring area were recorded using the testXpert II software. The curve shown in Figure 5 was selected for further analysis.

### 4.2. Biaxial Tensile Test

The biaxial tensile tests were carried out on a test stand consisting of a dynamic testing machine MTS 319.25, equipped with a special fixture enabling the performance of biaxial tensile tests on a uniaxial machine and the Aramis 3D system with a v2017 camera used to measure displacement [35]. Cross-shaped samples with the shape and dimensions shown in Figure 6 and Figure 7 were used for the tests. 

The use of the ARAMIS system requires the preparation of samples by painting their surfaces. Thanks to this, the sample surface is scanned by the optical system and on this basis its image is created along with the displacement values.

As in the case of uniaxial stretching, the biaxial tensile tests were carried out within the range of engineering strain ε = 0–60%. During the test, the axial force was measured on the testing machine, the displacement of the testing machine beam and the deformation in one of the tensile directions. In the case of biaxial stretching, the value of the engineering strain can be calculated as:(21)$$\varepsilon =2\varepsilon_{1}=2\varepsilon_{2}=2\left(\frac{\Delta l}{l_{0}}\right)=2\left(\frac{l_{\max}-l_{0}}{l_{0}}\right)$$
where: *ε*_1_, *ε*_2_—deformation in directions 1 and 2, Δ*l*—elongation, *l*_max_—maximum length, *l*_0_—initial length. 

Each of the three samples was subjected to a full cycle of loading and unloading with the constant elongation of the sample Δ*l* = 4.8 mm (measurement base *l*_0_ = 16 mm, thickness *g*_0_ = 4 mm). In the case of biaxial tensile test, the value of the equivalent stress was calculated as:(22)$$\sigma =\frac{F\cdot tg\alpha }{2\cdot b_{0}\cdot g_{0}}$$
where: *F*—force value measured on a testing machine, *b*_0_—the initial width of the sample arm, *g*_0_—initial sample thickness, *α*-the angle of inclination of the oblique arm. 

An example of a sample during the biaxial tensile test and the Aramis device are shown in Figure 8. 

### 4.3. Plane Strain Tensile Test 

Tensile tests in a flat deformation state were carried out on a ZWICK/ROELL Z030 testing machine, equipped with an optical extensometer and testXpert II software. A view of the machine and the sample during the test are shown in Figure 9. Specimens with the following dimensions were used: width a = 250 mm, height b = 125 mm, thickness g_0_ = 4 mm. 

The samples were mounted in special holders, the sample’s gripping part was 50 mm long. The initial distance between the jaws was 25 mm.

As in the previous tests, the plane strain tensile tests were carried out in the range of engineering strain *ε* = 0 ÷ 60%. During the test, the force and elongation of the sample were measured. In the case of the plane strain tensile test, taking into account the condition of a constant volume of the deformed material, the engineering strain can be calculated as:(23)$$\varepsilon =\frac{\sqrt{12}}{3}\cdot \varepsilon_{1}=\frac{\sqrt{12}}{3}\cdot \left(\frac{\Delta l}{l_{0}}\right)$$
where: *ε*_1_—strain component in the tensile direction, Δ*l*—elongation, *l*_0_—initial length.

Each of the three samples was subjected to a full cycle of loading and unloading with the constant elongation of the sample Δ*l* = 5.2 mm (measurement base *l*_0_ = 10 mm, thickness *g*_0_ = 4mm). 

### 4.4. Stress–Strain Curve

A characteristic feature of elastomers is that their deformation is not directly proportional to the applied load, in other words, these materials exhibit non-linear properties. Additionally, in the case of elastomers, the stress–strain characteristic depends on the type of applied load [26]. Figure 10 shows the curve σ = f(ε) determined experimentally on the basis of the performed tests, where: σ—engineering stress, ε—engineering strain. The course of the individual curves confirms that the stress–strain characteristic of the tested elastomer, and thus its response to the applied load, depends on the load scheme. The highest stress occurs during the biaxial tensile test, and the lowest during the uniaxial tensile test. Moreover, the difference between the stress values in individual material tests increases as the deformation increases. 

## 5. Determination of the Constants in the Constitutive Equations

The knowledge of the coordinates of individual points of the experimentally determined stress–strain curve is the basis for determining the values of the coefficients (material constants) in the individual constitutive equations. In this work, MSC Marc/Mentat 2020 was used to determine the constants in the constitutive equations selected for the study. After entering the coordinates of the points of the stress–strain curve determined by the experimental tests, the software adjusts the regression curves of selected constitutive models with the experimentally determined curves. This adjustment can be made on the basis of data from one or more tests. Based on the fit of the curves from the individual tests, the software calculates the numerical values of the material constants for the individual constitutive equations. In this paper, the differential evolution algorithm was used to determine the coefficients in the constitutive equations. This algorithm was proposed in 1995 and is used for optimization in continuous spaces [36].

The values of the material constants were determined for six phenomenological models (see Table 1), the Ogden model with two function components (see Table 2) and two micro-mechanical models (see Table 3). In each of these models, the material constants were calculated for variants I–IV. In variants I–III, the material constants in the investigated constitutive equations were determined on the basis of the results of only one material test, i.e., uniaxial tensile test (variant I), biaxial tensile test (variant II) or plain strain tensile test (variant III). However, in variant IV, the material constants were determined on the basis of the results of all three material tests. In addition, the research also included the Marlow model, the application of which requires the direct introduction of the appropriate stress–strain characteristics of the tested elastomer to the FEM software. Therefore, this model is not included in any of Table 1, Table 2 and Table 3. 

## 6. Experimental and Numerical Compression Test

In order to assess the effectiveness of the investigated constitutive equations, an experimental test of compression of the elastomeric cylinder was carried out. The cylindrical compression test of the samples from the tested elastomer with an initial diameter d_0_ = 20.5 mm and height of h_0_ = 32 mm was carried out on the ZWICK/ROELL Z030 testing machine. In order to eliminate the influence of sliding friction in the upsetting test, sandpaper was used between the faces of the sample and the plates of the testing machine. As a result, constant friction conditions were obtained. Each of the three tested samples was subjected to a single loading and unloading while maintaining constant deformation ε_h_ = 40% and the deformation was calculated as:(24)$$\varepsilon_{h}=\frac{h_{\max}}{h_{0}}\cdot 100\%$$
where: *h*_max_—maximum displacement, *h*_0_—initial sample height. 

The deformation *ε*_h_ was determined on the basis of the numerical modeling results in such a way that the deformation in the central area of the compressed sample had a value of about *ε* ≈ 0.6, and thus close to the deformation in the tests for determining the material constants (Figure 11).

During the tests, the compressive force and displacement were recorded. The thus obtained courses of the compression force of the elastomeric sample as a function of displacement (P = f(h)) and the compression force of the elastomeric sample as a function of deformation (P = f(*ε*_h_)) were used to verify the results of the numerical modeling. 

The numerical model of compression of an elastomer cylinder was created on the basis of the physical (experimental) model, i.e., with the same geometrical dimensions and boundary conditions, including contact conditions. A flat model was selected and analyzed with the assumption of the presence of an axially symmetric state of stress. In this case, the use of a two-dimensional model creates more opportunities as to the scope of the research carried out, e.g., due to the lower demand for computing power. The geometric model of the deformable cylinder has been simplified to its longitudinal half-section with the assumption that, in each longitudinal half-section of the deformed cylinder, the state of stress and deformation during its upset will be the same. In other words, the shape of the cylinder will change during the upset, but will remain an axisymmetric figure. The numerical model of compression of an elastomeric cylinder was created at MSC Marc/Mentat with a visible finite element mesh shown in Figure 12a. It consists of a longitudinal half-section of a deformable cylinder and non-deformable tools, i.e., a fixed bottom plate and a movable top plate. A “glue” type of contact was adopted between the elastomeric cylinder and the tool surfaces, which eliminated inaccuracies and errors related to the influence of sliding friction. A glue condition suppresses all relative motions between bodies through tyings or boundary conditions, applying them to all displacement degrees of freedom of the nodes in contact. For elements with nodes that also have rotational degrees of freedom, the rotations may be additionally constrained to provide a moment carrying glue capability. For the case that a connection is made to the face of a solid element, the rotations of the touching node are connected to the translations of the nodes of the contacted patch by a constraint relation based upon the large rotation RBE3 theory.

Type 82 elements were used. Element type 82 is a four-node, isoparametric arbitrary quadrilateral written for axisymmetric incompressible applications. As this element uses bilinear interpolation functions, the strains tend to be constant throughout the element [27]. The displacement formulation has been modified using the Herrmann variational principle. The pressure field is constant in this element. This element is preferred over higher-order elements when used in a contact analysis. The stiffness of this element is formed using four-point Gaussian integration. It can be used for either small strain behavior or large strain behavior. This element can be used in conjunction with other elements such as type 10 when other material behavior, such as plasticity, must be represented. 

The initial size of the finite elements was approx. 0.25 mm. Due to the fact that, during the upset without slip, large deformations occur, and thus large non-linearities of a geometric nature, local remeshing was applied in areas where the deformation of the elements was the highest. The equivalent deformation was adopted as the criterion for densifying the mesh. If the value of the equivalent deformation in an element located in a given area exceeded 0.25, then the mesh in the area of this element was compacted. The size of the elements after compacting the mesh was approx. 0.16 mm. An example of a model during compression with a locally compacted mesh is shown in Figure 12b. On the other hand, Figure 12c presents an example distribution of the Huber–Mises equivalent stresses for the strain *ε*_h_ = 40% using the 10th model and the variant II of determining material constants. Due to the contact conditions and deformation of the material, the stress distribution on the sample cross-section is non-uniform. The numerical model prepared this way was used to carry out numerical tests. Calculations for the compression process were carried out using individual constitutive models and the four variants for determining the constants in these models presented in Table 1, Table 2 and Table 3. Moreover, calculations were performed using the Marlow model, which was defined in MSC Marc by introducing the experimentally determined stress–strain curve. 

As a result of the numerical simulations for individual models and variants, the compression force as a function of displacement P = f(h) and the compression force as a function of deformation P = f(ε_h_) were obtained. The calculated P = f(ε_h_) curves for individual models and variants for determining the constants in the constitutive equations were compared with the experimental curves and are presented in Figure 13, Figure 14, Figure 15, Figure 16, Figure 17, Figure 18, Figure 19, Figure 20, Figure 21 and Figure 22. 

## 7. Evaluation of the Compression Test Modeling Effectiveness

It should be emphasized that it is practically impossible to satisfactorily and unequivocally assess the effectiveness of numerical modeling with the use of individual constitutive equations only on the basis of a visual comparison of the curves (Figure 13, Figure 14, Figure 15, Figure 16, Figure 17, Figure 18, Figure 19, Figure 20, Figure 21 and Figure 22). An efficient assessment of the effectiveness of numerical modeling using the investigated constitutive equations and four variants for determining the material constants in these equations is possible when one parameter is analyzed, which is a characteristic reference point in all analyzed cases. For this reason, in order to assess the effectiveness of the modeling of the elastomer compression test, the convergence indicator *ψ* was introduced, which allows us to determine the convergence of the calculated FEM force course with the experimentally determined force course. The numerical values of the indicator *ψ* for individual variants were calculated on the basis of the work done by both the calculated and experimental forces as: (25)$$\psi =\frac{W_{FEM}-W_{EXP}}{W_{EXP}}\cdot 100\%$$
where: $W_{FEM}=\int_{0}^{h_{\max}}F_{FEM}dh$ [N mm]—work done by the calculated compressive force, $F_{FEM}$, $W_{EXP}=\int_{0}^{h_{\max}}F_{EXP}dh$ [N mm]—work done by the compressive force in the experiment $F_{EXP}$.

In practice, the calculated numerical values of the *ψ* indicator may fall into three ranges. If:*ψ* = 0—force–displacement curves (calculated and experimental) coincide, which means the complete convergence of the results of FEM calculations with the experiment;*ψ* > 0—there is a discrepancy between the force–displacement curves, and the value of the calculated force is greater than the value of the experimental force;*ψ* < 0—there is a discrepancy in the course of the force–displacement curve, and the calculated force values are smaller than the force values in the experiment.

The smaller the absolute value of the *ψ* indicator, the greater the numerical model convergence with the experiment.

Figure 23, Figure 24, Figure 25 and Figure 26 show the calculated values of the *ψ* indicator for the investigated constitutive equations and individual variants for determining the material constants in these equations. Analyzing variant I, in which the material constants were determined only on the basis of the uniaxial tensile test, it can be seen that the convergence of the numerical modeling results and the experimental results largely depends on the model used (Figure 23). In this case, five of the tested models showed a very large overestimation of the results. In the case of models 2, 3, 4 and 6, *ψ* = 80.9%, while for model 7 it was the highest and amounted to approximately *ψ* = 116%. For the remaining models, an underestimation of the results was observed. Model 10 was the best in variant I, which showed the greatest convergence of the results with the experiment, with a slight underestimation (*ψ* = −5.5%). The analysis of the *ψ* indicator in variant II, in which the material constants were determined only on the basis of the biaxial tensile test, showed that the convergence of the results with the experiment to a small extent depends on the model used, while all the tested models showed underestimation of the results (Figure 24). In the case of six models (1−6), the same numerical value of the indicator *ψ* (*ψ* = −9.2%) was observed. In the case of model 7, this value was very similar and amounted to *ψ* = −9.0%. Model 8, for which *ψ* = −9.5%, showed a slightly greater discrepancy in the results. The largest discrepancy in this comparison (*ψ* = −12.9%) was demonstrated by model 9, and the smallest by model 10, for which *ψ* = −4.4%. On the other hand, on the basis of the analysis of the *ψ* value in variant III, in which the material constants were determined only on the basis of the plane strain tensile test, it can be concluded that the obtained convergence of the results with the experiment also depends on the model used. In this variant (Figure 25), two models showed overestimation of the results, i.e., model 3 (*ψ* = 12.6%) and model 4 (*ψ* = 9.5%). The greatest underestimation was shown by model 9 (*ψ* = −18.1%) and model 1 and model 5 (*ψ* = −17.4%). Model 2 (*ψ* = −8.0%) was much better. One can get the impression that the models 6 (*ψ* = −2.4%), 7 and 8, for which the coefficient *ψ* was 0.1% fared best in this comparison. It should be noted, however, that in the case of the latter models, such a low *ψ* indicator results from the specific course of the calculated force–deformation curve, which to some extent lies below the experimental curve (is underestimated), and to some extent lies above the experimental characteristics (is overestimated). Such curves are visible in Figure 18, Figure 19 and Figure 20, where the calculated forces (green line) intersect with the experimental forces (black line). It should be emphasized that, in such cases, the low value of *ψ* does not mean that the calculated curve and the experimental curve are almost completely convergent, but it is a result of the fact that these forces perform almost the same work. For this reason, in a few cases in which the calculated and experimental curves intersect (in such a way that there are similarly sized underestimated and overestimated areas), the indicator *ψ* expressed by Equation (25) is not reliable. Similarly to variant II (Figure 24), an analysis of *ψ* in variant IV, in which the constants in the constitutive equations were determined on the basis of three tests, showed that the convergence of the results with the experiment to a small extent depends on the model used, while all the tested models showed an underestimation of the results (Figure 26). Model 10 is not included in this comparison because its application is not possible on the basis of the results of several material tests at the same time, but only on the basis of the results of one selected material test. Among the nine tested models in variant IV, the highest convergence with the experiment was achieved by model 8 (*ψ* = −7.9%). Model 9 (*ψ* = −7.9%) turned out to be the least accurate in this comparison. The remaining seven models 1–6 and 8 fell within the range of *ψ* = (−11.7 to −11.5)%.

## 8. Conclusions

The aim of the work was to evaluate the effectiveness of ten selected constitutive equations for hyperelastic bodies in the numerical modeling of elastomer compression. Four variants for determining the material constants in the constitutive equations were used.

In conclusion, the research showed that determining the material constants of an elastomer for modeling its behavior under compressive load can be best achieved through the biaxial tensile test (variant II). This approach yielded modeling results that were independent of the constitutive equation or had minimal dependence, with a high convergence with the experimental data. The average convergence is around 10%. The most accurate model was found to be model number 10 (Marlow). Variant IV, which is more complex and involves the use of three tests to determine the material constants, was found to be less effective than variant II.

However, using variant I, which solely relied on the uniaxial tensile test, proved to be a more practical solution in engineering practice. The effectiveness of this approach depended on the constitutive equation used, and only a few models achieved a high convergence with the experimental data. Model 10 (Marlow) was again the best model in this variant, with a convergence of −5.5%. 

Overall, the research findings provide valuable insights into the accurate modeling of elastomer behavior, with practical implications for engineering applications.

## Figures and Tables

**Figure 1 materials-16-04121-f001:**
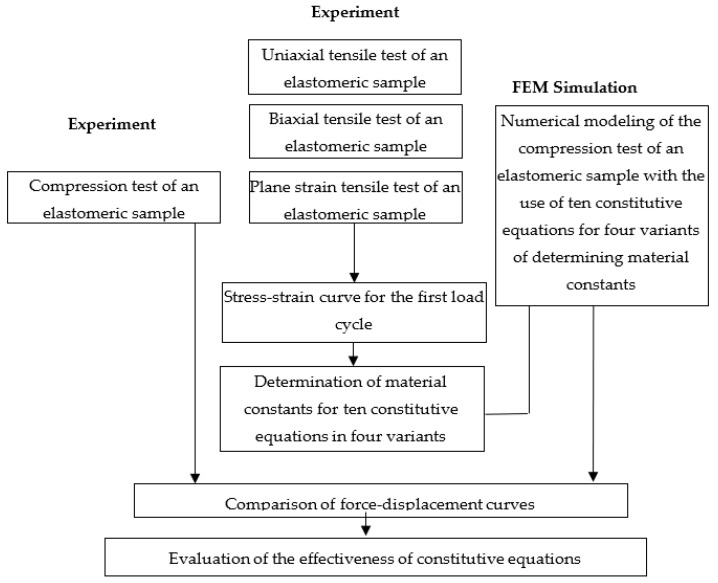
Scheme of the research carried out.

**Figure 2 materials-16-04121-f002:**
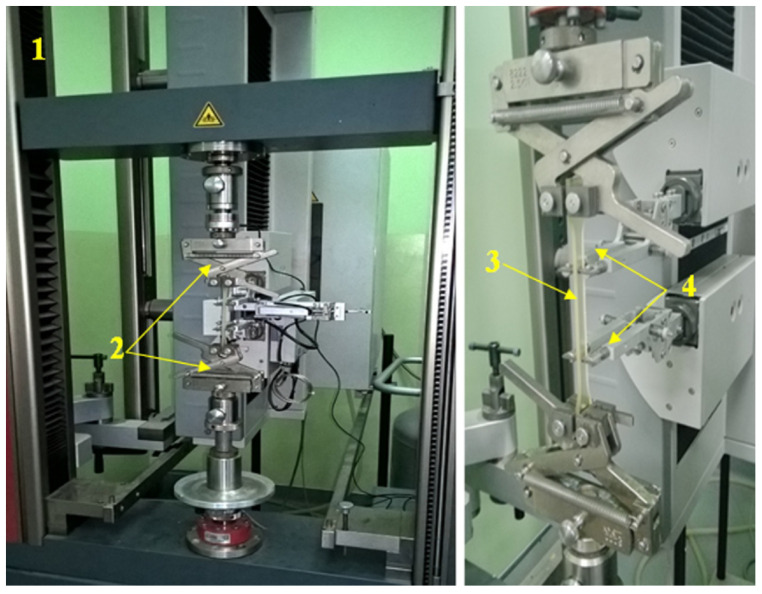
Uniaxial tensile test, view of the sample during the test: 1—testing machine, 2—fixtures, 3—specimen, 4—multiextensometer.

**Figure 3 materials-16-04121-f003:**
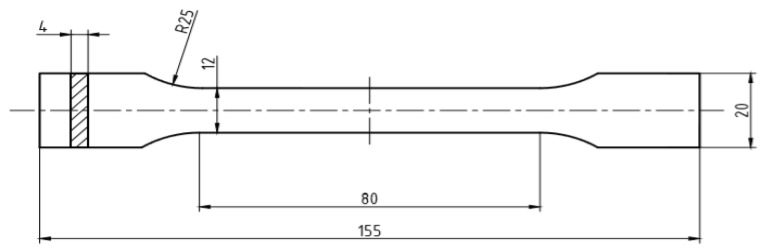
Shape and dimensions of samples for uniaxial tensile test, mm.

**Figure 4 materials-16-04121-f004:**
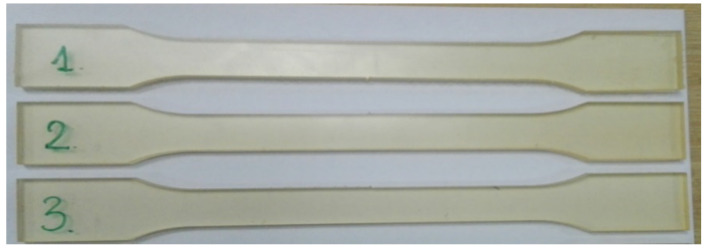
Elastomeric samples for uniaxial tensile test.

**Figure 5 materials-16-04121-f005:**
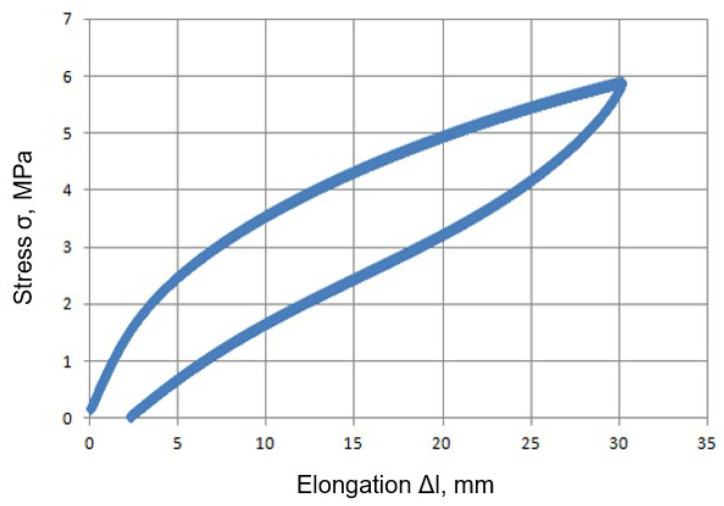
Stress as a function of elongation during loading and unloading of a sample in the range of deformation up to ε = 60% (σ—engineering stress calculated as the quotient of the tensile force to the initial cross-sectional area of the specimen in the measurement area, Δl—elongation in the measuring area).

**Figure 6 materials-16-04121-f006:**
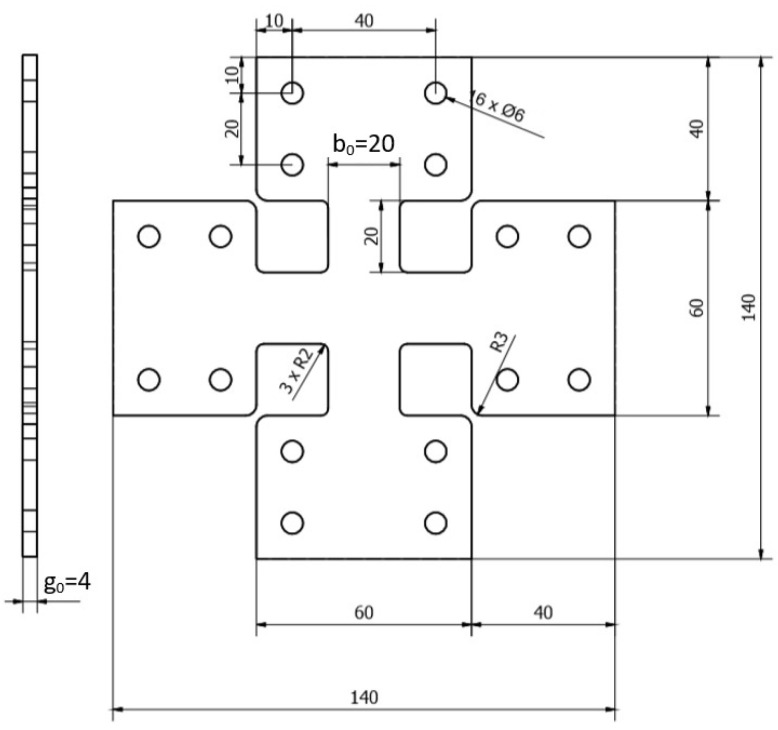
Dimensions of an elastomeric specimen for the biaxial tensile test [mm].

**Figure 7 materials-16-04121-f007:**
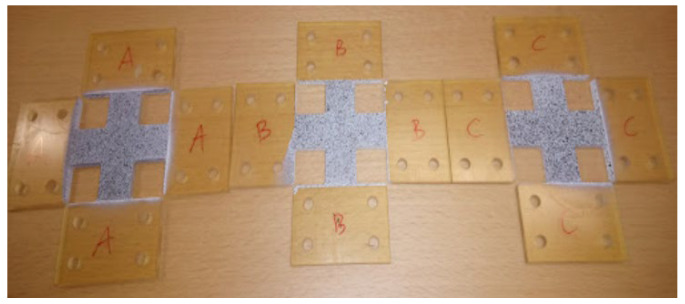
Elastomer samples prepared for the biaxial tensile test.

**Figure 8 materials-16-04121-f008:**
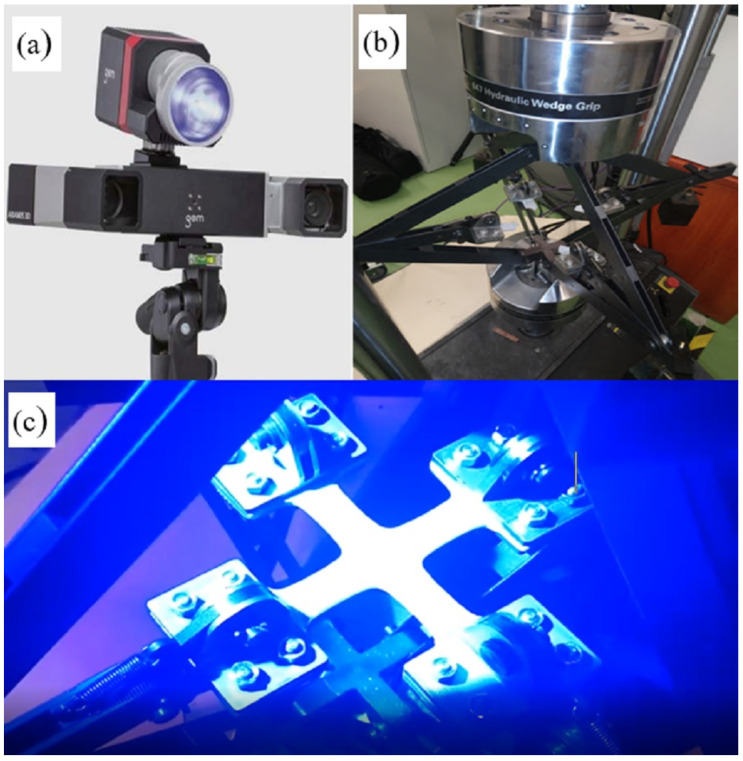
Biaxial tensile test: (**a**) ARAMIS 3D device, (**b**) biaxial tensile test fixture, (**c**) the sample during the test.

**Figure 9 materials-16-04121-f009:**
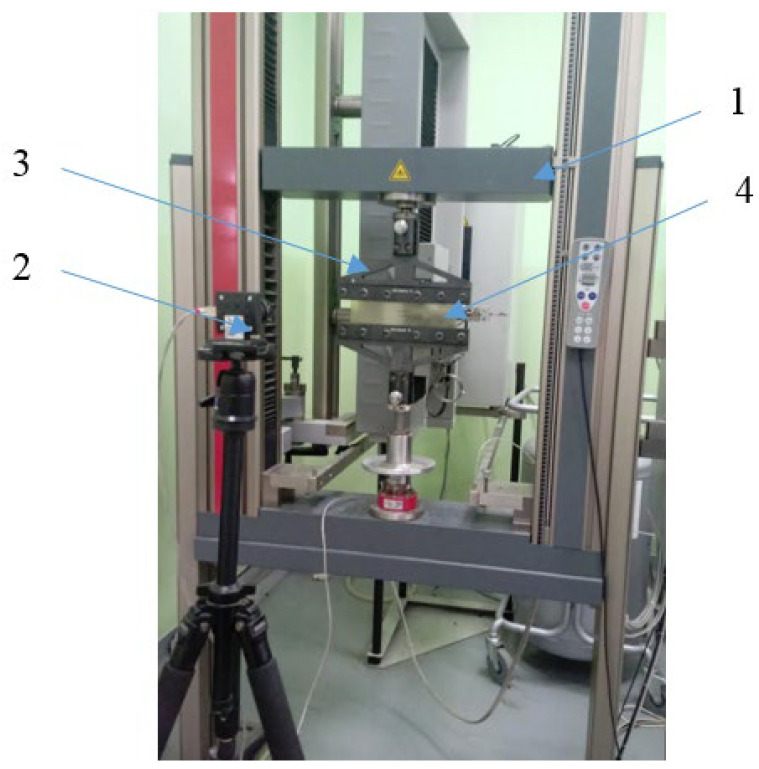
Plane strain tensile test: 1—testing machine, 2—optical extensometer, 3—clamping jaws, 4—sample.

**Figure 10 materials-16-04121-f010:**
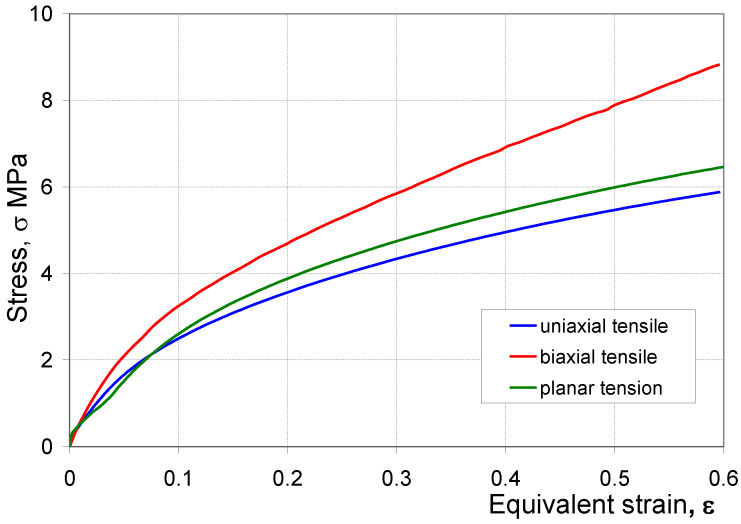
Experimental stress–strain curve for the three load patterns.

**Figure 11 materials-16-04121-f011:**
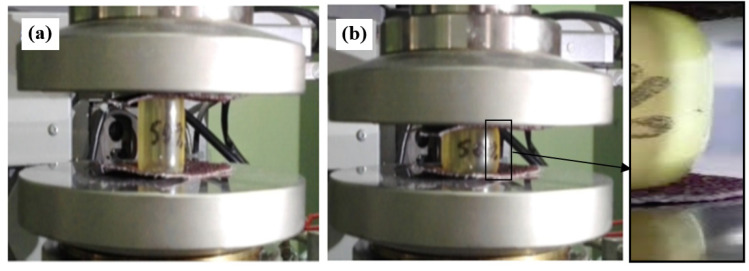
View of an exemplary sample in the initial (**a**) and final (**b**) compression stage.

**Figure 12 materials-16-04121-f012:**
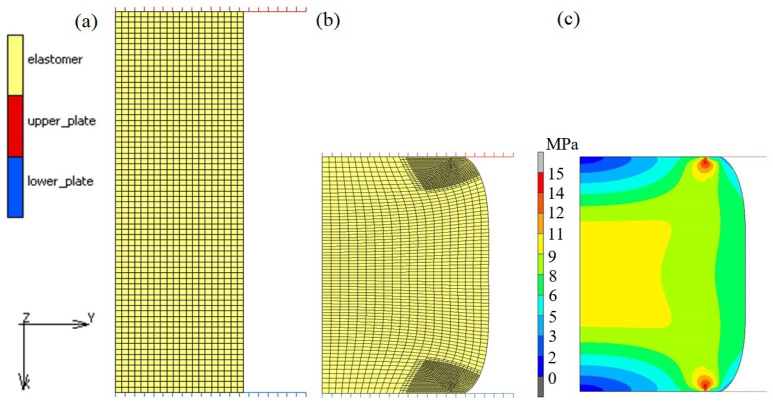
Numerical model of the cylinder compression process: (**a**) initial model, (**b**) model during loading, (**c**) exemplary Huber–Mises stress distribution (model 10, variant II, ε_h_ = 40%).

**Figure 13 materials-16-04121-f013:**
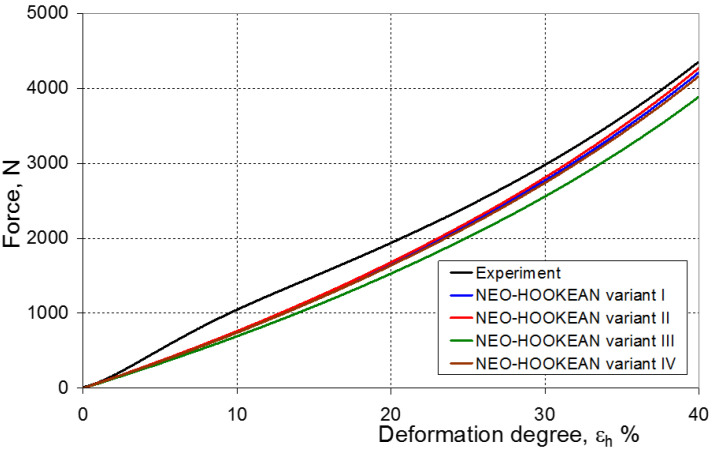
Experimental and calculated elastomer compression curves using the Neo-Hookean model.

**Figure 14 materials-16-04121-f014:**
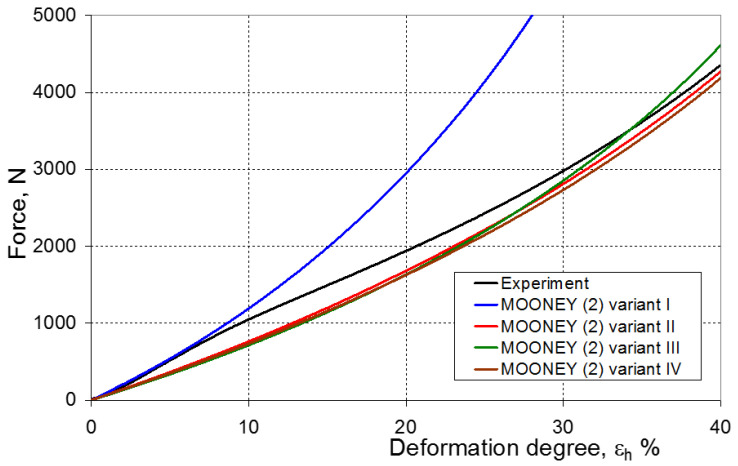
Experimental and calculated elastomer compression curves using the Mooney (2) model.

**Figure 15 materials-16-04121-f015:**
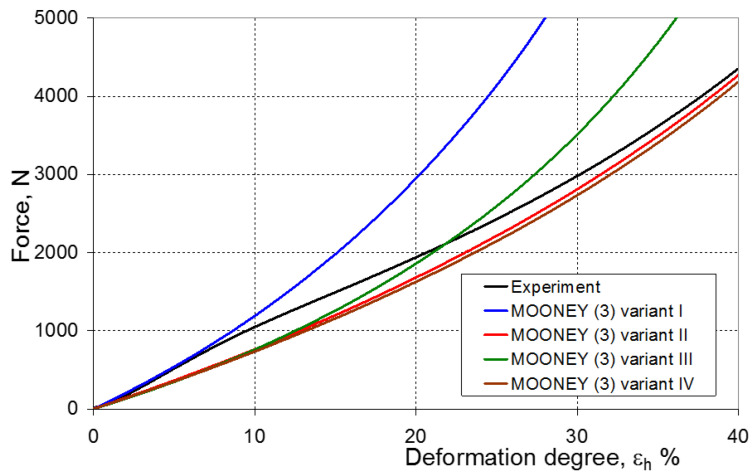
Experimental and calculated elastomer compression curves using the Mooney (3) model.

**Figure 16 materials-16-04121-f016:**
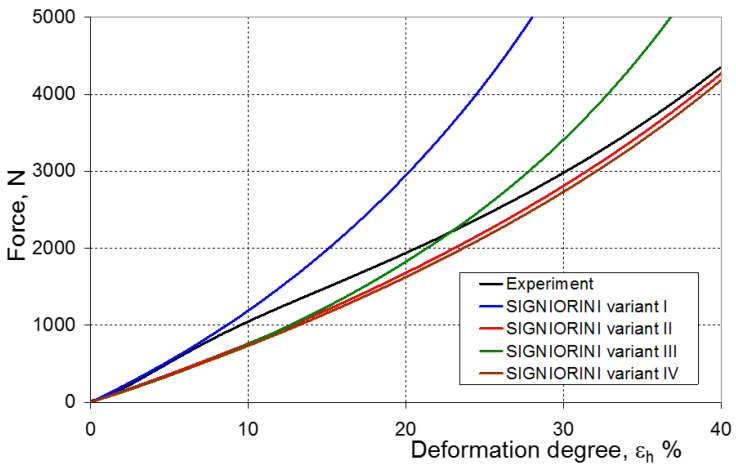
Experimental and calculated elastomer compression curves using the Signiorini model.

**Figure 17 materials-16-04121-f017:**
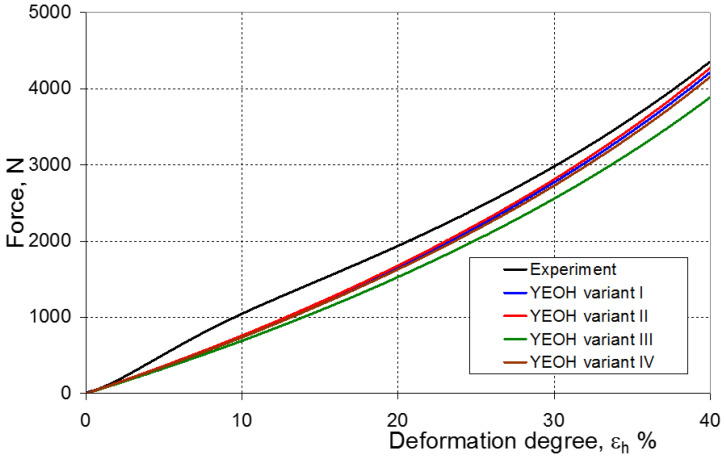
Experimental and calculated elastomer compression curves using the Yeoh model.

**Figure 18 materials-16-04121-f018:**
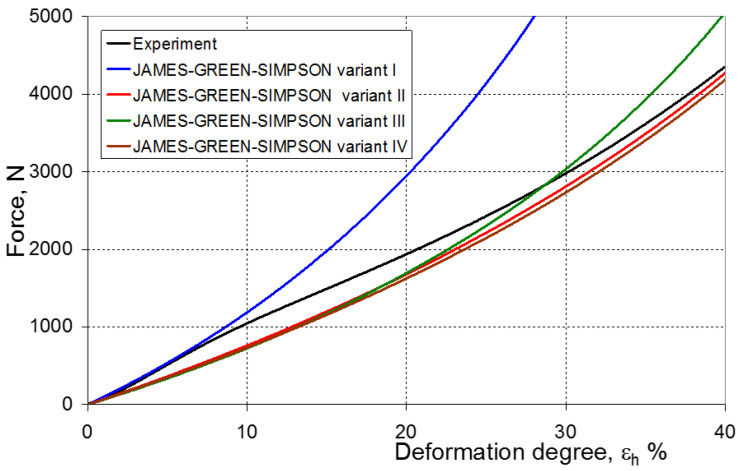
Experimental and calculated elastomer compression curves using the James–Green–Simpson model.

**Figure 19 materials-16-04121-f019:**
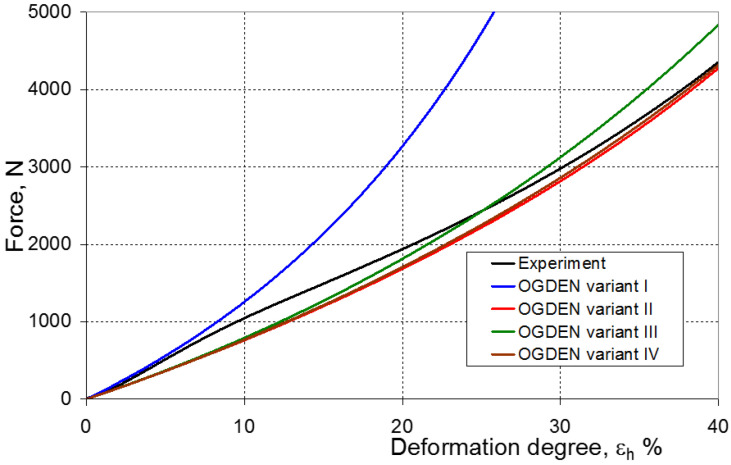
Experimental and calculated elastomer compression curves using the Ogden model.

**Figure 20 materials-16-04121-f020:**
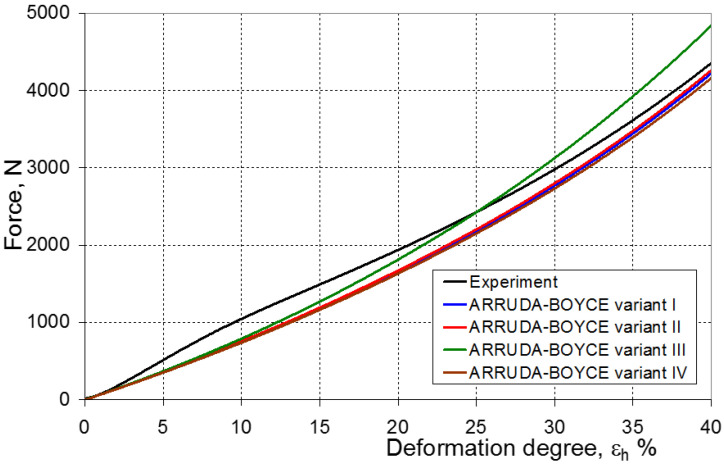
Experimental and calculated elastomer compression curves using the Arruda–Boyce model.

**Figure 21 materials-16-04121-f021:**
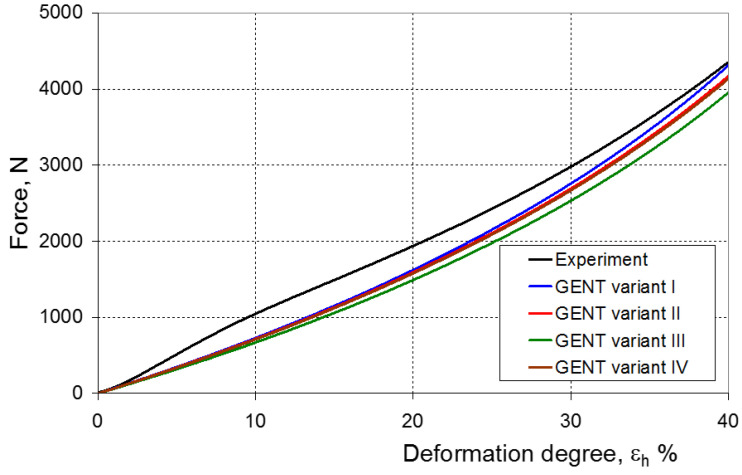
Experimental and calculated elastomer compression curves using the Gent model.

**Figure 22 materials-16-04121-f022:**
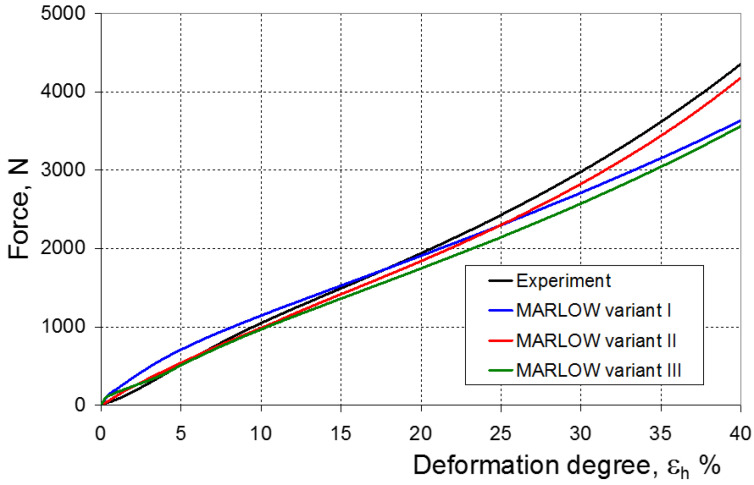
Experimental and calculated elastomer compression curves using the Marlow model.

**Figure 23 materials-16-04121-f023:**
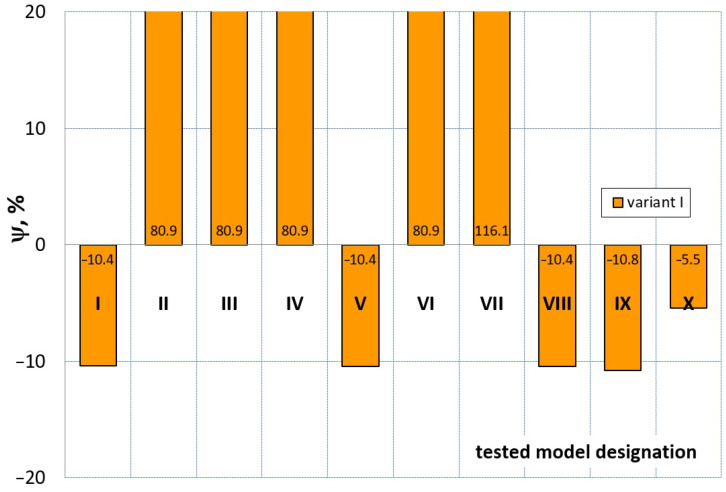
Distribution of the *Ψ* indicator for the tested models (1–10) using variant I for determining the material constants.

**Figure 24 materials-16-04121-f024:**
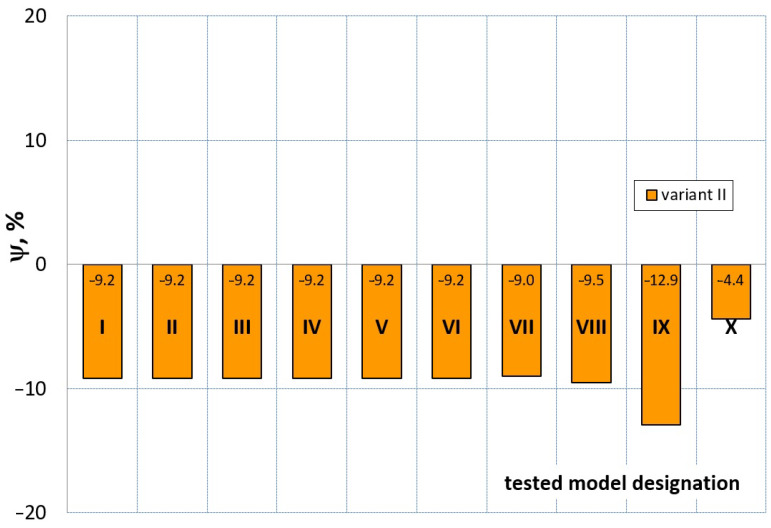
Distribution of the *Ψ* indicator for the tested models (1–10) using variant II for determining the material constants.

**Figure 25 materials-16-04121-f025:**
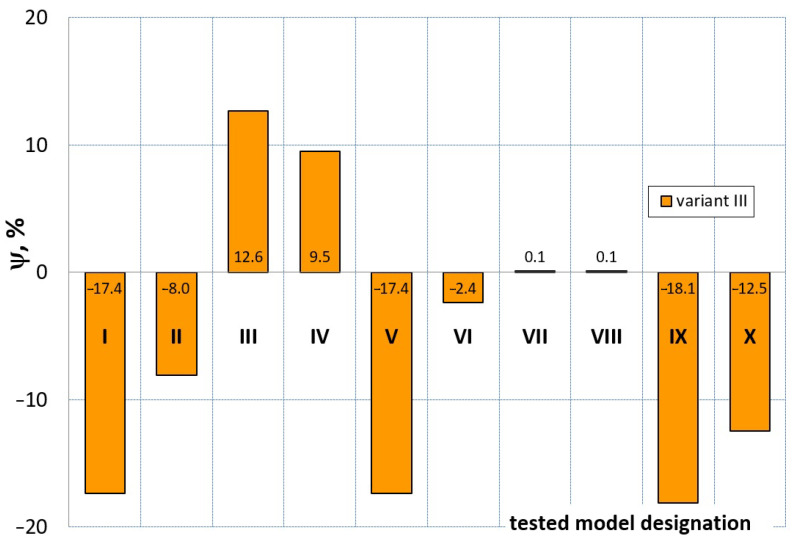
Distribution of the *Ψ* indicator for the tested models (1–10) using variant III for determining the material constants.

**Figure 26 materials-16-04121-f026:**
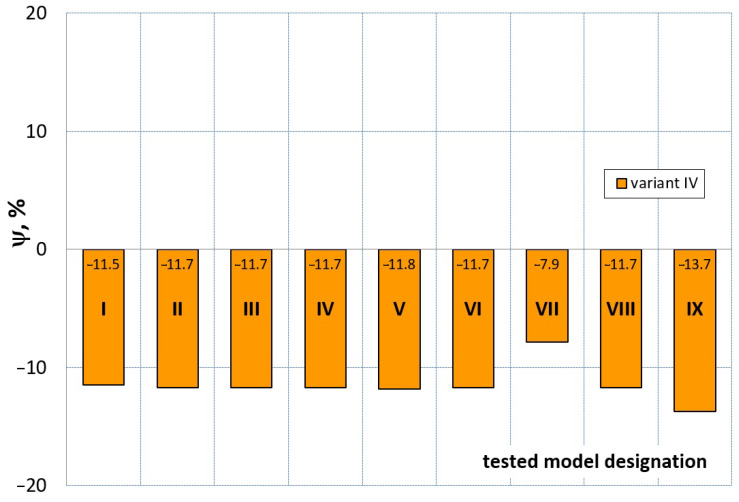
Distribution of the *Ψ* indicator for the tested models (1–9) using variant IV for determining the material constants.

**Table 1 materials-16-04121-t001:** Material constants determined for variants I–IV for models in the first sample load cycle up to 60% deformation.

No	Model	Variant I–IV	Material Constants					Coeficient of Determination
			C_10_	C_01_	C_11_	C_20_	C_30_	R^2^
1	NEO-HOOKEAN	I	2.872	-	-	-	-	0.987
		II	2.912	-	-	-	-	0.996
		III	2.65					0.993
		IV	2.838	-	-	-	-	0.992
2	MOONEY (2)	I	7.521 × 10^−13^	4.097	-	-	-	0.993
		II	2.912	2.147 × 10^−8^	-	-	-	0.996
		III	1.938	0.712				0.993
		IV	2.717	0.080	-	-	-	0.992
3	MOONEY (3)	I	1.047 × 10^−8^	4.096	1.586 × 10^−9^	-	-	0.993
		II	2.912	2.803 × 10^−8^	1.462 × 10^−9^	-	-	0.996
		III	0.340	2.310	1.686 × 10^−8^			0.993
		IV	2.717	0.080	3.860 × 10^−8^	-	-	0.992
4	SIGNIORINI	I	1.314 × 10^−6^	4.097	-	7.065 × 10^−8^	-	0.993
		II	2.912	6.276 × 10^−8^	-	3.643 × 10^−8^	-	0.996
		III	0.585	2.065	-	5.271 × 10^−9^	-	0.993
		IV	2.718	0.079	-	2.456 × 10^−8^	-	0.992
5	YEOH	I	2.872	-	-	1.419 × 10^−9^	2.767 × 10^−9^	0.987
		II	2.912	-	-	5.538 × 10^−10^	2.893 × 10^−10^	0.996
		III	2.65	-	-	2.002 × 10^−8^	1.048 × 10^−7^	0.993
		IV	2.825	-	-	1.148 × 10^−8^	0.003	0.992
6	JAMES–GREEN–SIMPSON	I	1.304 × 10^−7^	4.096	2.339 × 10^−7^	1.565 × 10^−7^	2.006 × 10^−8^	0.993
		II	2.912	4.574 × 10^−8^	5.843 × 10^−8^	1.378 × 10^−8^	2.359 × 10^−7^	0.996
		III	1.500	1.150	1.292 × 10^−8^	1.378 × 10^−8^	1.317 × 10^−8^	0.993
		IV	2.717	0.080	2.302 × 10^−12^	1.624 × 10^−12^	2.085 × 10^−7^	0.992

**Table 2 materials-16-04121-t002:** Material constants determined for variants I–IV for the Ogden model in the first cycle of loading the sample up to 60% deformation.

No	Model	Variant I–IV	The Number of Components of the Function N	Material Constants		Coeficient of Determination
				µ_n_	α_n_	R^2^
7	OGDEN	I	2	−5.809	−2.896	0.993
				4.762 × 10^−5^	0.054	
		II		−1.104 × 10^−5^	−0.142	0.996
				6.033	1.931	
		III		289.102	0.040	0.995
				0.055	0.044	
		IV		−1.440 × 10^−4^	−4.11 × 10^−2^	0.992
				8.416	1.376	
				8.416	1.376	
				8.416	1.376	
				8.416	1.376	

**Table 3 materials-16-04121-t003:** Material constants determined for variants I–IV for micro-mechanical models in the first sample load cycle up to 60% deformation.

No	Model	Variant I–IV	Material Constants		Coeficient of Determination
8	ARRUDA–BOYCE		NKΘ	N	R^2^
		I	5.638	37.966	0.987
		II	5.727	52.440	0.996
		III	5.204	39.525	0.993
		IV	5.589	52.4399	0.992
9	GENT		E	I_M	
		I	16.688	18.998	0.986
		II	16.433	26.220	0.995
		III	15.344	19.762	0.992
		IV	16.284	26.220	0.991

## Data Availability

The data presented in this study are available on request.

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
