# Peer review of "Modeling Elastomer Compression: Exploring Ten Constitutive Equations"

_materials, 2023, doi:10.3390/ma16114121_

Round 1

Reviewer 1 Report

1.      The authors are suggested to describe a little about polymer physics/ chemistry in the introduction part.

2.      Line 128, this is not linear elastic shear modulus. It is initial shear modulus. Authors are suggested to read the following publication and try to compare “G” or “E” obtained using different models. 

3.      Authors are suggested to discuss finite element modeling in brief/ The type of elements, boundary conditions should be elaborated. Authors can refer to: https://doi.org/10.1016/j.polymertesting.2020.106856

4.      The stress-strain equations for uniaxial and biaxial cases should be elaborated. 

5.      Authors should compare the curve fitting quantitatively i.e., in terms of R2.

6.      Title needs a major revision. “Ten constitutive equations should be eliminated.” Try to make it a little generalized.

7.      Figure 12 should show distribution of stresses.

English can be improved and typos should be checked thoroughly

Reviewer 2 Report

PFA

Minor grammar and English correction is required.

Round 2

Reviewer 1 Report

1. Authors should discuss types of elements and other informations related to FEA.

2. FEM modeling should not be used as 'M' in FEM already denotes modeling.

3. R-square should be denoted for each model in table.

4. As similar studies have been reported a decade ago, authors should discuss novelty.

Minor english editing required.
